Natural Marine Cloud Brightening in the Southern Ocean
Gerald G. Mace[1], Sally Benson[1], Ruhi Humphries[2,3], Peter M. Gombert[1], Elizabeth
Sterner[1]
[1]Department of Atmospheric Sciences, University of Utah, Salt Lake City, Utah
[2]Climate Science Centre, CSIRO Oceans and Atmosphere, Melbourne, Australia
[3]Australian Antarctic Program Partnership, Institute for Marine and Antarctic Studies,
University of Tasmania, Hobart, Tasmania, Australia
Corresponding Author Information:
Gerald "Jay" Mace, Professor
Department of Atmospheric Sciences, University of Utah
135 South 1460 East Rm 819 (819 WBB)
Salt Lake City, Utah, 84112-0110
Cell Phone:  801 201 7944
Office Phone: 801 585 9489
Email: jay.mace@utah.edu
Fax:  801 860 0381

Abstract: The number of cloud droplets per unit volume ($N_d$) is a fundamentally important property of marine boundary layer (MBL) liquid clouds that, at constant liquid water path, exerts considerable controls on albedo. Past work has shown that regional $N_d$ has direct correlation to marine primary productivity (PP) because of the role of seasonally varying biogenically-derived precursor gasses in modulating secondary aerosol properties. These linkages are thought to be observable over the high latitude oceans where strong seasonal variability in aerosol and meteorology covary in mostly pristine environments. Here, we examine $N_d$ variability derived from five years of MODIS level-2 derived cloud properties in a broad region of the summer Eastern Southern Ocean and adjacent marginal seas. We demonstrate latitudinal, longitudinal, and temporal gradients in $N_d$ that are strongly correlated with the passage of air masses over high PP waters that are mostly concentrated along the Antarctic Shelf poleward of 60°S. We find that the albedo of MBL clouds in the latitudes south of 60°S is significantly higher than similar LWP clouds north of this latitude.

Short Summary: The number cloud droplets per unit volume is a significantly important property of clouds that controls their reflective properties. Computer models of the Earth's atmosphere and climate have low skill at predicting the reflective properties of Southern Ocean clouds. Here we investigate the properties of those clouds using satellite data and find that the cloud droplet number in the Southern Ocean is related to the oceanic phytoplankton abundance near Antarctica and cause clouds there to be significantly brighter than clouds further north.

1. Introduction

The cloud and precipitation properties of the Southern Ocean (SO) have received considerable attention since Trenberth and Fasullo (2010) identified a high bias in surface-absorbed solar energy there (McFarquhar et al., 2020). This bias has been traced to erroneously small Marine Boundary Layer (MBL) cloud cover in simulations of the Southern Ocean climate (Bodas-Salcedo, et al., 2016; Naud et al., 2016). The actual SO cloud climatology and associated albedo are dominated by geometrically thin MBL clouds (Mace et al., 2010; Mace et al., 2020, 2021). Because the predominant shallow boundary layer clouds rarely precipitate (Huang et al., 2016), they are sensitive to cloud condensation nuclei (CCN) concentrations (Twohy and Anderson, 2008; Painemal et al., 2017).

In the SO, the CCN seasonal cycle (Ayers and Gras, 1991; Vallina et al. 2006; Gras and Keywood, 2017) is reflected in basin-wide cloud property variations (Krüger and Graßl, 2011). McCoy et al. (2015) and Mace and Avey (2017) also found that MODIS- and A-Train-derived cloud properties over the SO, demonstrate a similar seasonal cycle in cloud droplet number concentration ($N_d$) as for CCN. The basin wide variability in CCN and cloud albedo have been shown to be correlated with marine primary productivity (PP – defined as the net organic matter, mostly produced by phytoplankton, that is suspended in the ocean; Vallina et al., 2006; Krüger and Graßl,2011; McCoy et al., 2015). McCoy et al. (2020) argue that the SO can be viewed as an analog of the preindustrial Earth. Given the large natural seasonal variability in CCN and clouds, the

SO is a natural laboratory to understand the processes that contribute to simulated
aerosol-related indirect forcing variability in climate models (Carslaw et al. 2013).
CCN and cloud droplet $N_d$ in the SO are higher in Summer when significant latitudinal
gradients have been documented in the SO Australasian sector (Humphries et al.,
2021).  Using time of flight aerosol chemical speciation monitor (ACSM) and ion

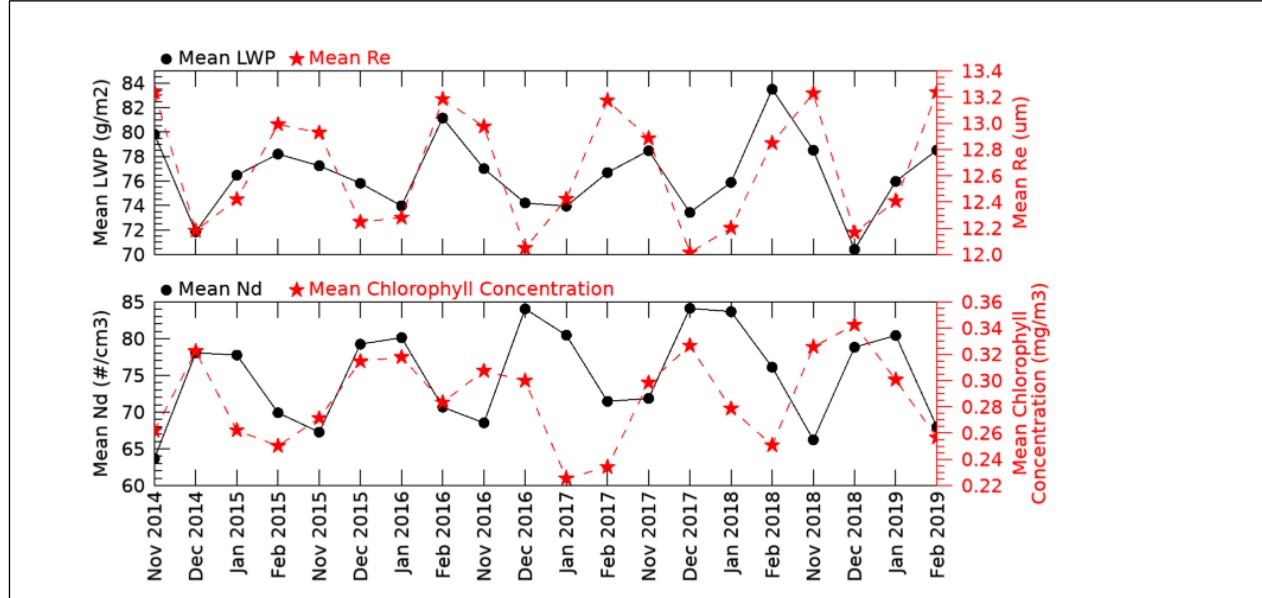

Figure 1. Monthly-averaged cloud properties and Chlorophyll-a (Chl-a) derived from MODIS data over the analysis domain. Top Panel: LWP (black dots and solid line) and effective radius (Re, red star and dashed line).  Bottom Panel: $N_d$ (black dots and solid line) and Chlorophyl Concentration (red star and dashed line).

concentrations from filter samples, Humphries et al., (2021) analyzed the covariance of
aerosol chemistry, CCN at 0.5% supersaturation, and Condensation Nuclei (CN) larger
than 10 nm collected aboard Australian research vessels during the 2018 Austral
Summer (McFarquhar et al., 2021). While sulfates were a major compositional
component of aerosol at all latitudes during summer these compounds were in higher
fractional abundance poleward of 65°S where overall CCN numbers were higher by
~50%. Chloride derived from sea salt was dominant in the region equatorward of 65°S
but was mostly absent south of 65°S.   The ratio of CCN to CN at 0.5% supersaturation
increased considerably south of 65°S suggesting unique aerosol chemical processes
compared to the open ocean.  Humphries et al. (2021) also discusses how this
compositional boundary in aerosol chemistry is often very distinct in the East Antarctic
waters between 60°S and 65°S.  Following Humphries et al. we will refer to this belt as
the Atmosphere Compositional Front of Antarctica (ACFA).   Humphries et al. (2021)
conclude that aerosol, newly condensed from gas phase sulfur species such as from
the oxidation of dimethyl sulfide (DMS), are an important component of high latitude
CCN.  These products of phytoplankton physiology are released into the atmosphere
from the highly productive waters from ~60°S to the Antarctic – a region well known for
a vast marine food web (Deppler and Davidson, 2017; Behrenfeld et al., 2016).
Mace et al. (2021a) derived $N_d$ and other cloud microphysical properties from non-
precipitating stratocumulus clouds using shipborne remote sensing data.  They found
that stratiform clouds poleward of the ACFA had significantly higher $N_d$ than
equatorward.  One particular case took place when the Icebreaker Aurora Australis was
at the Davis Antarctic station just east of Prydz Bay (~77°E) between 1 and 5 January
2018 and featured nearly continuous high $N_d$ clouds (> 150 cm$^{-3}$) occurring in a
southerly flow passing over the ship that had trajectories from the Antarctic Continent.
Similarly, Twohy et al., (2021) report that the highest concentrations of aerosol
composed primarily of non-sea salt sulfates in the free troposphere north of 60°S
observed from research aircraft in Summer 2018 had occurred in airmasses that had
originated recently from over the Antarctic continent. See also Shaw et al. (1988) for an
early examination of the role of biogenic sulfate in modulating summertime aerosol
along coastal Antarctica. Shaw et al. (2007) expands on this idea as does Korhonen et
al., (2008).
2.  Results

See Appendix A for methods and definitions.  Approximately 40,000 1° latitude by 2°
longitude MBL cloud scenes per month meet our criteria for liquid phase non
precipitating clouds in the analysis domain.  This number varies by ~25% in a seasonal
cycle that is due mostly to our solar zenith angle criteria.  A seasonal cycle is evident in
the monthly-averaged cloud properties. LWP and $r_e$ have seasonal minima in the
months of December and January.  Due to an $r_e^{-5/2}$ dependence, $N_d$ is of opposite phase

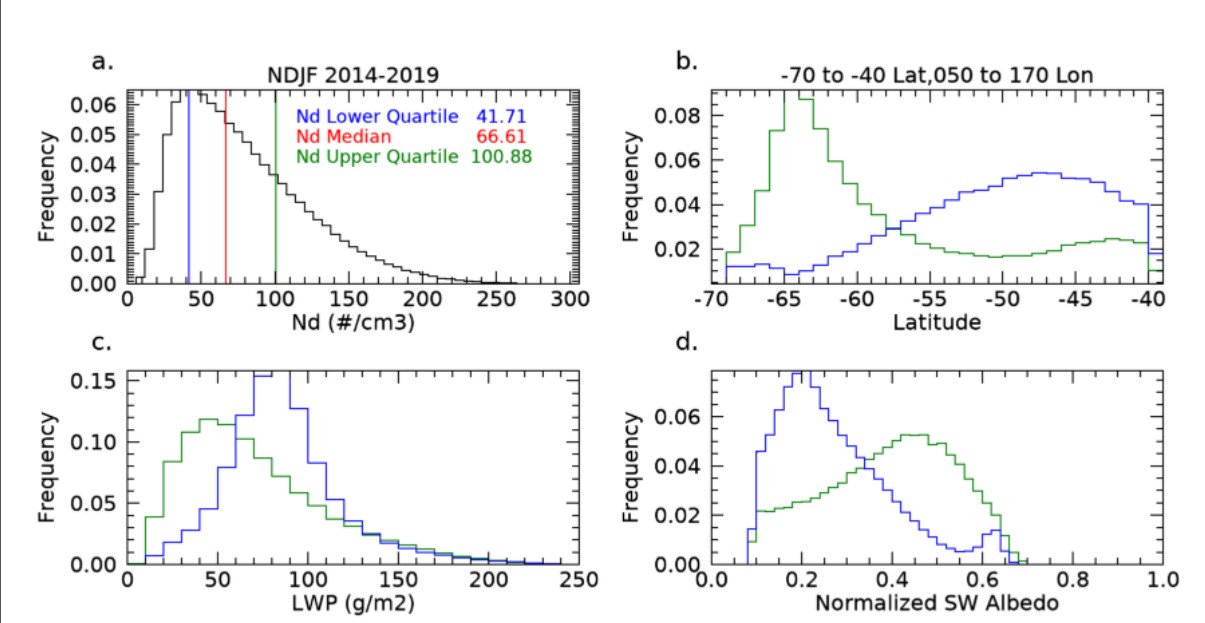

Figure 2.  a) $N_d$ frequency distribution from the cloud scenes in the analysis domain during
the 5-years of summer months analyzed.  Vertical lines are defined in the inset.  b) The
latitudinal distribution of the cloud scenes that compose the high and low $N_d$ quartiles.  c)
the distributions of liquid water path for the high and low Nd quartiles, d) the distribution of
normalized CERES solar albedo of the high and low Nd quartiles.   The normalization
procedure is described in the appendix.  The colors of the histograms in paneles b, c, and d,
are as described in the inset of panel a.

with $r_e$ and correlated with it at -0.93.  The seasonal variability in LWP ($r_e$) is on the
order of 7% (4%) and is small in comparison to Nd (~25%).   $\tau$ and $r_e$ are derived from
the visible and near infrared reflectances with the MODIS level 2 retrieval algorithm
(Nakajima and King, 1990).  LWP is, then, derived from
$$\tau = \frac{3}{2\rho_w}\frac{LWP}{r_e}$$      Equation 1.
that is derived in Stephens (1978). It is reasonable to consider whether seasonal
variations in $N_d$, perhaps linked to CCN, might be associated with variability in LWP.
We find that LWP decreases as $N_d$ increases with a correlation coefficient in the
monthly means of -0.60 in the monthly means.

In four of the five years, we see by inspection of Figure 1 that Chl-a leads changes in $N_d$
by approximately 1 month. The correlation coefficient of $N_d$ and Chl-a increases from
0.27 to 0.60 when $N_d$ is lagged from 0 to 1 month in the Figure 1 time series although
this result should be interpreted with caution given the break between February and
November in the time series. These results are broadly like those presented by McCoy
et al., (2015) and Mace and Avey (2017). McCoy et al. (2015) link $N_d$ variations to PP
using regression analysis of MODIS derived $N_d$ against a biogeochemical
parameterization of biogenic sulfate and organic mass fraction (See also Lana et al.,
139     2012).
We find a broad distribution of scene-averaged $N_d$ (Figure 2a) with median, lower and
upper quartile values of 66 cm$^{-3}$, 42 cm$^{-3}$ and 101 cm$^{-3}$ respectively. Henceforth, we
focus our analysis on the groups of scenes that are less than and greater than the
upper and lower quartiles. The high and low $N_d$ scenes have distinct latitudinal
occurrence distributions (Figure 2b) with low $N_d$ scenes peaking broadly at 48°S while
the high $N_d$ scenes demonstrate a modal occurrence near 64°S. Overall, the $N_d$ gradient
implied by Figure 2 is correlated with the latitudinal distribution of imager-derived Chl-a
(i.e., Deppler and Davidson, 2017). The seasonally averaged $N_d$ gradient is also
discussed in McCoy et al., (2020). Differentiating seasonally varying properties north
and south of the ACFA (not shown), we find a clear differentiation in $r_e$ and $N_d$ with
smaller $r_e$ south of the ACFA (mean $r_e$~11um, $N_d$ ~100) compared to north (mean
$r_e$~13um, $N_d$ ~67 cm$^{-3}$). LWP is slightly larger by ~7% south of the ACFA. Both
regions have a distinct seasonal cycle in cloud properties shown in Figure 1 although
the southern latitudes have larger interannual variability likely owing to variations in
annual sea ice extent and melt. The LWP distribution of the high $N_d$ quartile is
significantly shifted to lower values compared to the low $N_d$ quartile LWP distribution
(Figure 2c). This finding is in accordance with the observational and theoretical work
presented in Glassmeier et al., (2021) who argue that closed cell stratocumulus that
dominate the clouds examined here have increased entrainment drying under higher $N_d$
conditions. Figure 2c and 2d illustrate that even though the high $N_d$ quartile scenes tend
to have lower LWP, their solar albedo ($A$) tends to be significantly higher than the low
$N_d$ quartile scenes illustrating the influence of cloud microphysics on the radiative
forcing of these clouds.
The high $N_d$ scenes occur predominantly poleward of the ACFA (Figure 3). Interestingly
we find that the latitudinal gradient weakens slightly west of 90°E with a broad region of
higher $N_d$ occurrence in the vicinity of the Kerguelen Rise where PP is higher (Cavagna
et al., 2015). Establishing causality between regions of high PP and cloud properties is
challenging (i.e., Meskhidze and Nenes, 2006; Miller and Yuter, 2008). While we find
seasonal associations over broad regions here, the chain of causality between
phytoplankton and clouds is not immediate or even necessarily direct because the
chemical processes take time to evolve and can move along chemical pathways that
have divergent outcomes (Woodhouse et al., 2013). To increase cloud $N_d$, new CCN
must be formed. Formation of new CCN can occur when sulfur compounds emitted
from the ocean surface nucleate after oxidation in the presence of sunlight. This
process of new particle formation occurs in the absence of other aerosol and often
requires mixing of the gaseous compounds from the boundary layer into the low-aerosol
free-troposphere where the newly formed aerosol can be transported widely (Shaw,
2007; Korhonen et al., 2008).  Other pathways are possible such as deposition of
sulfate compounds onto primary sea salt particles that modify the chemical properties of
existing CCN rather than nucleating new CCN (Fossum et al., 2020) or even removal of
sulfur compounds from the gas phase via aqueous phase oxidation in clouds
(Woodhouse et al., 2013).

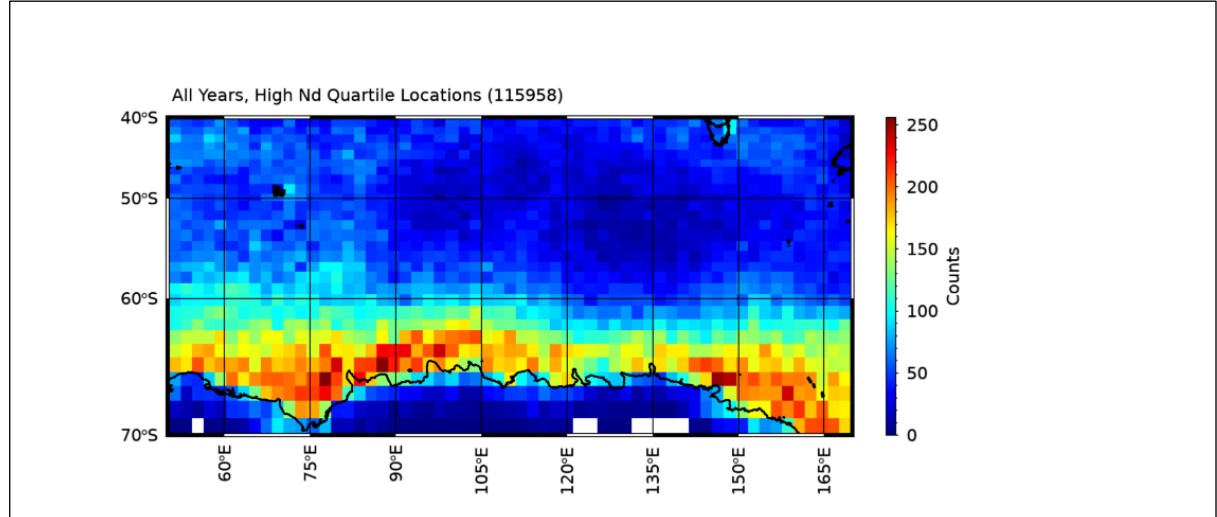

Figure 3.  Geographic distribution of the high $N_d$ quartile cloud scenes.  Number in parentheses show the total of number cloud scenes from the 5-year summer data set.

Given the foregoing discussion, it seems reasonable that an airmass that is producing
clouds with certain features could be interacting with an aerosol population that has
evolved over periods of days (Brechtel et al., 1998).   In addition, natural cloud
processes such as collision and coalescence of drops tend to cause $N_d$ to decrease
while precipitation efficiently scavenges CCN, thereby lowering CCN concentration and
even modifying their composition and size through aqueous processing (Hoppel et al.,
1986). With larger $r_e$ north of the ACFA, the collision-coalescence process is likely more
active (Freud and Rosenfeld, 2012) and could explain the latitudinal difference in
adiabaticity (see methods) found in in situ data.   For instance, Kang et al. (2022)
analyzed data collected from Macquarie Island (54.6°S, 158.9°E) and found that, not
only were most clouds drizzling, but that precipitation as light as 0.01 mm hr[-1] could
reduce $N_d$ by ~50%.  Therefore, a cloud field should be considered as the product of
both local dynamics and thermodynamics primarily with modulation by a local
population of CCN.  To examine the role of airmass history, we calculate the 5-day back
trajectories using the Hybrid Single-Particle Lagrangian Integrated Trajectory (HYSPLIT;
Stein et al., 2015) model using the Global Data Assimilation System (GDAS; Kamitsu,
1989) as input.   The parcel's endpoint is the central latitude and longitude of the cloud
scene, and the location and model output are stored hourly.

South of the ACFA, the histories of the populations tend to be statistically different
(Figure 4).  The low $N_d$ clouds are more likely to be observed in airmasses that have
trajectories that originated in the open ocean region to the north of the ACFA.  High $N_d$
scenes rarely evolve in airmasses that originate in the open ocean to the north of the
ACFA.  The likelihood is that an airmass that has produced a high $N_d$ cloud scene south
of the ACFA latitude has spent most of the previous 5 days over latitudes south of the
ACFA.  North of the ACFA, the latitude distributions during the months of November and
February (not shown) are essentially identical for the high and low Nd quartiles.
However, for December and January, we find that the high $N_d$ clouds observed north of
the ACFA have an increased likelihood of trajectories emanating from south of the
ACFA during the 5-days prior to the MODIS observation.
3.   Discussion and Conclusions
Using MODIS level 2 cloud property retrievals and the technique developed in
Grosvenor et al. (2018; hereafter G18) to estimate $N_d$, we examine the latitudinal and
seasonal cycles of non-precipitating liquid-phase clouds in the Australasian sector of the
Summertime Southern Ocean.  The $r_e$ and $N_d$ have distinctive differences north and
south of the ACFA but demonstrate similar seasonal cycles.  We infer that the spatial
and temporal variability in cloud $N_d$, and $r_e$ are at least partially a function of the
geographic and temporal variability in CCN that, in turn, is related to the seasonality of
primary sources such as sea salt and the latitudinal variability in marine PP.   The
highest $N_d$ clouds tend to be overwhelmingly found along the East Antarctic coastal
waters south of the ACFA.

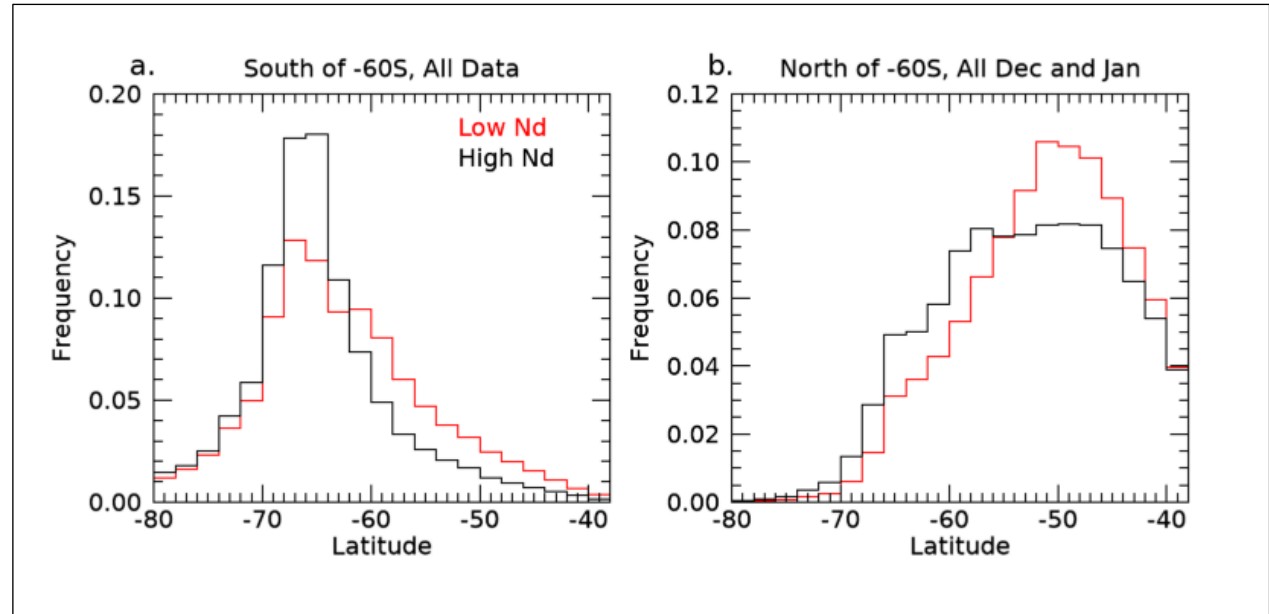

Figure 4. Distributions of the latitudes crossed by the 5-day back trajectories for the low (red) and high (black) $N_d$ cloud scenes.

Because aerosol precursor gasses like DMS often require trajectories through the free
troposphere to nucleate new particles that then take time to reach CCN sizes
(Korohonen et al., 2008; Shaw et al. 2007), we examine the back trajectories of the
airmasses observed with high and low $N_d$ south of the ACFA and find significant
differences. Low $N_d$ cloud scenes are more likely to have arrived south of the ACFA
from northerly trajectories that would have transported low CCN air dominated by sea
salt. The high $N_d$ cloud scenes are more likely to have trajectories that have remained
adjacent to or had passed over the Antarctic continent. North of the ACFA, while the
trajectory statistics for the high and low $N_d$ quartiles in November and February are
nearly identical, during December and January the high $N_d$ clouds scenes tend to have
an increased likelihood of arriving north of the ACFA from southerly trajectories,
suggesting that high CCN airmasses are being transported northward especially during
December and January.
Given that the main difference between the source regions north and south of the ACFA
is the magnitude of the marine PP, and given previous analyses of CCN compositional
sensitivity to marine biological factors (e.g. Humphries e al., 2021; Vallina et al., 2006;
Lana et al., 2012; McCoy et al, 2015), we conclude that the biological source of sulfate
precursor gasses and the slackening of surface winds with latitude during Summer
plays a dominating role in controlling the latitudinal gradients in the properties of weakly
precipitating MBL cloud fields over the Southern Ocean. Figure 5 summarizes our
findings by presenting composite seasonal cycles of MBL cloud scenes north and south
of 60°S. The LWP in both latitudinal bands go through a weak seasonal cycle. The
significant contrast in optical depth between the northern and southern bands is, we
infer, mostly caused by the latitudinal contrast in $N_d$. Based on available evidence, we
conclude that the differences in $r_e$ in MODIS retrievals are causally linked to oceanic PP
gradients that drive CCN, and thereby $N_d$, to be higher over the southern region.   This
sensitivity, in turn, plays a significant role in modulating the regional albedo (*A*) and,
thereby, influences the input of sunlight to the surface ocean.  We note that the
seasonal cycle in *A* is different between the northern and southern latitude domains (a
topic for future work), however, always A of the southern domain is higher than that of
the northern domain.   However, we should be careful not to overstate this case.  Cloud
processes that consume $N_d$ and modify CCN (i.e. precipitation and cloud processing)
also play a role in modulating cloud $N_d$ and therefore regional *A* (Kang et al., 2022;
McCoy et al., 2020).  The airmass history and source region, while apparently
important, are among many factors involved.
Since the magnitude of PP is significantly lower north of the ACFA throughout the
summer season, a similar seasonal cycle in $N_d$ and $r_e$ suggests that CCN derived from
DMS oxidation of precursor gasses emitted primarily from Antarctic coastal waters
perhaps seeds much of the rest of the Southern Ocean with biogenic sulfate aerosol as
observed in recent airborne observations (Twohy et al., 2021).   The northerly transport
of these high sulfate airmasses out of the Antarctic coastal waters (Figure 4b) and
southerly transport of low sulfate air masses into the Antarctic coastal region near the
surface (Figure 4a) have been reported by Humphries et al. (2016, 2021) and Shaw
(1988) and observed in the free troposphere with recent research aircraft
measurements (Twohy et al. 2021).
Our ability to identify natural marine cloud brightening (Latham et al., 2008) due to
aerosol-cloud coupling is a direct result of the absence of other anthropogenic and
continental influences in the pristine SO.   As argued by McCoy et al. (2020), it seems
clear that in several important ways, the Southern Ocean is the last vestige of the
preindustrial atmosphere allowing us to constrain processes that remain important to
our understanding of the global climate (Carslaw et al., 2013).

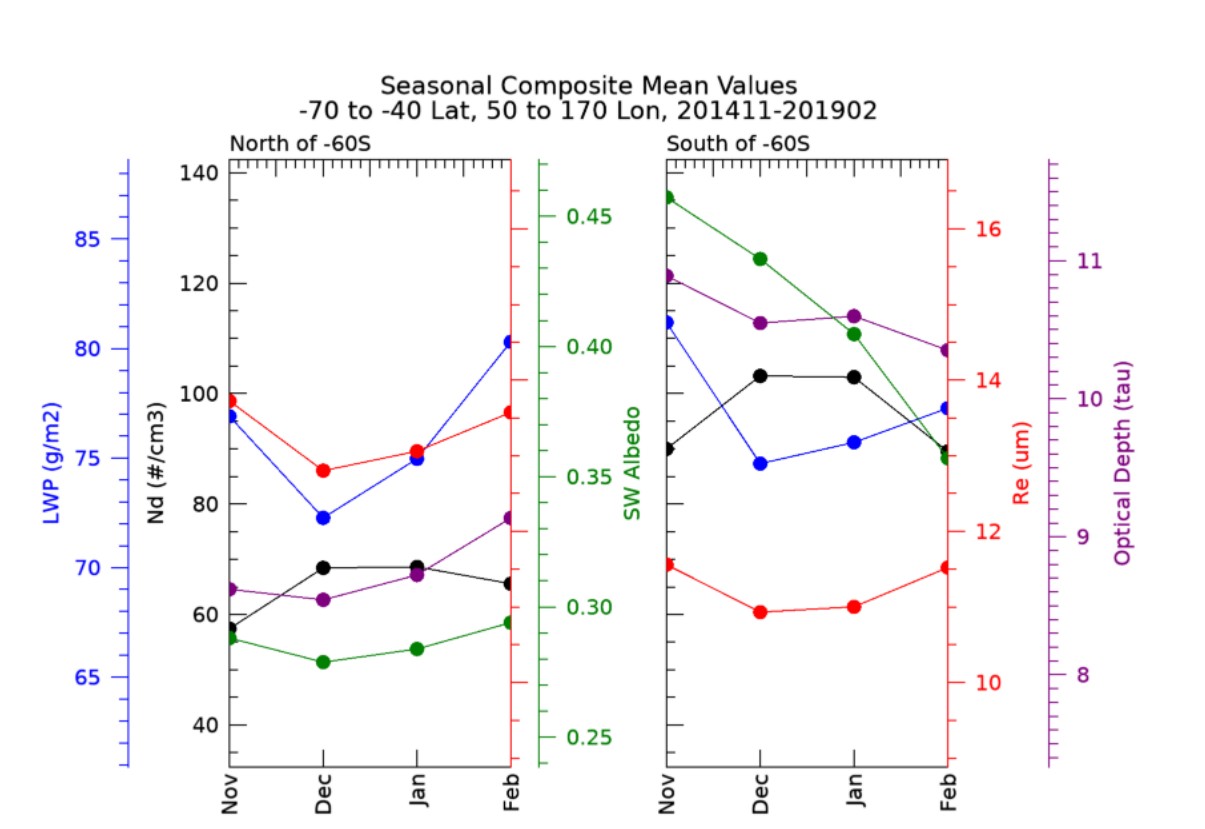

Figure 5.  Composite seasonal cycle of cloud properties.  Each data point is comprised of the monthly mean of cloud scenes in the analysis domain compiled November, 2014-February, 2019.  The effective radius (Re, red curve) and the optical depth (solid purple curve) are taken directly from MODIS Level-2 retrievals.  The liquid water path (LWP, blue curve) and cloud droplet number ($N_d$, black curve) are derived as described in the text.  The solar (SW) albedo (green curve) is derived from CERES data and normalized to a solar zenith angle of 45° as described in the Appendix.

Appendix.  Methods

We use MODIS imager-derived Level-2 retrievals (Platnick et al., 2015) of effective
radius ($r_e$) and optical depth ($\tau$) from five summer periods (2014-2019) collected
between the latitudes of 45°S and 76°S and longitudes of 40°E and 170°E to focus
roughly on where the ships and aircraft sampled in Summer 2017-18.  We calculate $N_d$
using the method derived and evaluated in G18:

$$N_d = \frac{\sqrt{5}}{2\pi\kappa}\left(\frac{f_{ad}c_w\tau}{Q_{ext}\rho_w r_e^5}\right)^{1/2}$$    (A1)

where $\rho_w$ is the density of liquid water (1 g cm$^{-3}$), $f_{ad}$ is an adiabaticity assumption, $c_w$ is
the vertical derivative of the adiabatic liquid water content, $Q_{ext}$ is the extinction efficiency
that is typically assumed to be 2 for cloud droplets, and $\kappa$ is the cubed ratio of $r_e$ to $r_v$.  As
noted by G18, $N_d$ depends on $r_e^{-5/2}$, which implies that the sensitivity or the rate of change
of $N_d$ to retrieved $r_e$ goes as the -7/2 exponent.  Any biases in $r_e$, then would significantly
bias $N_d$.  G18 provide a thorough evaluation of the sources of uncertainty in $N_d$ due to
assumptions of adiabaticity, scene heterogeneity, etc., and conclude that $N_d$ derived
using equation 1 applied to MODIS cloud retrievals has an overall uncertainty of ~80%.
The most uncertain quantity in the assumptions used in Equation A1 is $f_{ad}$ since the cloud
vertical structure is not constrained by MODIS measurements. Using cloud thickness from
ship-based cloud radar and lidar along with retrieved LWP from collocated microwave
radiometer (Mace et al., 2021a), we estimate the value of $f_{ad}$ in nonprecipitating
stratocumulus observed during the summer of 2018 (McFarquhar et al., 2021).  We find
that the mean and standard deviation of $f_{ad}$ north of the ACFA is 0.66 and 0.48,
respectively.  South of the ACFA, the mean and standard deviation of $f_{ad}$ is 0.93 and 0.60,
respectively. For the calculations of $N_d$ in equation A1, we use a constant value for $f_{ad}$ of
0.8.  $N_d$ is proportional to the square root of $f_{ad}$ , therefore, $\frac{\partial ln N_d}{\partial \ln f_{ad}} = \frac{1}{2}$ and a fractional
variation in $f_{ad}$ on the order of 0.5 would imply an uncertainty in $N_d$ of 0.25.  Furthermore,
we expect in regions with $f_{ad}$ higher (lower) than 0.8 the $N_d$ would be biased low (high).
As we show, the regions with higher $N_d$ tend be in the south and lower $N_d$ in the north
counter to these expected biases.  Additionally in this study, we will be examining
differences in spatially averaged $N_d$ that are greater than a factor of 2.   These results
imply that bias and random error due to uncertainty in $f_{ad}$ is unlikely to significantly
influence the qualitative findings of this study.
Another source of systematic bias could be from the quantity $\kappa$ that can be shown to be
a function of the variance of the droplet size distribution and is assumed to be a constant
at 0.7.  G18 discusses this issue in some detail and concludes that there may be
systematic biases on the order of 12% that could be a function of $N_d$ in pristine conditions.
While this quantity can be investigated with data collected in situ, no such data exists in
stratocumulus clouds south of the ACFA.  Therefore, we recognize a potential source of
bias due to $\kappa$ that is likely much smaller than the systematic latitudinal differences we find.
Given the uncertainties in $N_d$ at the pixel level, we implement a filtering and averaging
scheme to focus on liquid phase, weakly precipitating cloud scenes.  We define a scene
as a 1° latitude by 2° longitude domain where pixels are reported in the MODIS L2 data
to be of liquid-phase. We assume that clouds are weakly precipitating clouds if the cloud
liquid water path (LWP) < 300 g m$^{-2}$.  We require that the sensor and solar zenith angles
($\theta$)  at that pixel are less than 30° and 60°, respectively. The maximum $\theta$ requirement is
motivated by the findings of Grosvenor and Wood (2014) who find that systematic errors
in MODIS retrievals increase significantly for $\theta$>60°. The $\theta$ requirement causes us to
focus on the months from November through February.  We require at least 1000 1-km
resolution pixels with these characteristics to exist within a scene (typical number
>10000).  In addition, we require that no more than 10% of the pixels have a cloud top
temperature less than -20°C to ensure the absence of ice phase hydrometeors.  Cloud
properties within a scene are averaged.

Collocated cloud albedos (*A*) of the cloud scenes are analyzed.  *A* is derived from the
Clouds and the Earth's Radiant Energy System (CERES) Energy Balanced and Filled
(EBAF) version 4.0 (Loeb et al, 2018) data collected using instruments on board Aqua
and Terra.  The albedo is derived by dividing the upwelling shortwave flux at the top of
the atmosphere (TOA) by the downwelling shortwave flux at TOA.  Because *A* has a
solar zenith angle dependence, (Minnis et al. 1998), we normalize all albedo values to
$\theta$=45° (approximately the mean value of $\theta$ for the analysis domain and months
analyzed) with an empirical method using theoretically calculated *A* ($\hat{A}$) as a function of
latitude presented in Minnis et al. (1998 – their figure 7).  The normalization is
implemented by first approximating the latitudinal dependence of *A* for various cloud
optical depths ($\tau$) using the following regression equation: $\hat{A} = 0.51 - 0.43\mu_0^{1/2} +$
$0.17\ln\tau$  where $\mu_0 = \cos\theta$.  $\hat{A}$ approximates the variation of A with latitude within ~15%
at $\tau$=8.  The fit decreases in accuracy at higher and lower $\tau$ increasing to an uncertainty
of ~30% for $\tau$=2 and $\tau$=32 (these values of $\tau$ (2, 8, 32) are those presented in Minnis et
al., 1998, Figure 7). The averaged $\tau$ of the MBL cloud scenes in our analysis is
approximately between 9 and 11 (Figure 5) so we expect that $\hat{A}$ is typically a reasonable
approximation of *A*.   The normalization of all *A* to $\theta = 45°$ is accomplished by
multiplying the CERES *A* by the ratio $\frac{\hat{A}(\mu_0(\theta=45),\tau)}{\hat{A}(\mu_0,\tau)}$ where $\tau$ is from the MODIS cloud
scene. The magnitude of the ratio applied to the data ranges from 0.85 at higher
latitudes to 1.2 at lower latitudes with an average near 1.
Author Contributions:  GM led the overall conception, data analysis of the study and
interpretation of the results.  SB was responsible for implementing data analysis code
and generation of figures.  RH provided background on aerosol and provided insight
regrading various aspects of the study.  MPG and ES assisted GM in the study design
and implementation.
Competing Interests:  The authors declare no conflict of interest.

Acknowledgements:  This work was supported by NASA Grant 80NSSC21k1969 and
DOE ASR Grants DE-SC00222001 and DE-SC0018995.  All data used in this study are
available in public archives.   Computer code for this study including all analysis code
and graphic generation code is written in the IDL language.  Code is available upon
request to the corresponding author.

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
