# Peer review of "Natural Marine Cloud Brightening in the Southern Ocean"

_Atmospheric Chemistry and Physics, 2022_

## Referee Comment (RC2)

**Review of "Natural Marine Cloud Brightening in the Southern Ocean" by Mace et al. (acp-2022-571)**

The presented study analyzes strong gradients in the cloud droplet concentration found in the Southern Ocean, using five years of satellite observations. The authors show that these stark differences in the cloud microphysical composition can be traced back to biological primary production at the Antarctic Shelf, from where airmasses with high cloud droplet concentrations are moved to the north, while low cloud droplet concentrations originate from the open sea equatorward.

Despite many technical issues and a few minor comments, I enjoyed reading this manuscript. In a concise way, the article informs about the aerosol and cloud microphysics of an important region of the Earth. Thus, I support publication in *Atmospheric Chemistry and Physics* once my concerns are addressed. However, this article is also submitted to be published as an *Atmospheric Chemistry and Physics Letter*. In the current form, I cannot support the publication in this format, as I will outline in my only major comment below, but I am willing to be convinced otherwise.

**Major Comment**

*Does the article meet the requirements for an Atmospheric Chemistry and Physics Letter?* It is stated that an *Atmospheric Chemistry and Physics Letter* must fulfill the following requirements (see www.atmospheric-chemistry-and-physics.net/about/manuscript_types/acp_letters.html):

- Important discoveries and research highlights in atmospheric chemistry and physics.

- Solutions to or progress with long-standing and important questions in atmospheric research.

In its current form, the manuscript does not give substantial hints on how these requirements are fulfilled. I understand that the Southern Ocean is one of the least understood parts of the climate system, with inherent problems in modeling it. But how does the presented study contribute to improving its understanding? What are the important discoveries made? While I admit that the presented research is very interesting, the authors should use the opportunity to frame their work and highlight the advancements made through their work.

**Minor Comments**

Ll. 47 – 50: What cloud process is sensible to the CCN concentration? In non-precipitating clouds, an increase in CCN will not change the precipitation efficiency. I guess the authors refer to the cloud albedo.

Ll. 56 – 58: Define what "primary production" is. This will also help to frame the importance of the work (major comment). Do ll. 79 – 84 also refer to primary production?

Ll. 105 – 108: What is the purpose of this sentence? $N_d$ and $r_e$ are in opposite phase, not out of phase. So there is a very tight relationship.

L. 108: Replace "cycle" with "variability"?

Fig. 2b: Add labels. The blue line represents locations with $N_d > 101$ cm$^{-3}$ and the green line $N_d < 42$ cm$^{-3}$. What about adding a red line for $42$ cm$^{-3} < N_d < 101$ cm$^{-3}$?

Ll. 113 – 117: The LWP changes non-monotonically with $N_d$. It increases with $N_d$ for precipitation stratocumulus, but decreases for non-precipitating stratocumulus due to increasing entrainment rates (Glassmeier et al. 2021).

Ll. 131 – 134: Why is $N_d$ only large at the ACFA and not below it?

Ll. 135 – 139: With a mean $r_e$ of 13 µm north of the ACFA, droplet coalescence might decrease $N_d$. See, e.g., Freud and Rosenfeld (2012), who showed that at 14 µm surface precipitation occurs, i.e., there is probably some drizzle at a slightly smaller $r_e$. This should be discussed.

Ll. 275 – 278: The significantly lower adiabaticity north of the ACFA could be due to precipitation, triggering the transition of closed- to open-cell stratocumulus. Discuss the possibility of precipitation.

Ll. 297 – 300: Clouds with $r_e$ > 14 μm are usually precipitating (Freud and Rosenfeld 2012). Declaring all pixels with $r_e$ < 50 μm to non-precipitating clouds will cause a substantial bias. Please elaborate.

**Technical Comments**

L. 32: "Both" usually refers to two objects. Here, it refers to three (latitudinal, longitudinal, and temporal). Revise.

L. 45: SO for Southern Ocean is already defined. Use it.

L. 55: To what is "respectively" referring to?

L. 66: Why is "TOF" defined? It is never used.

L. 88: "ACF" is not defined. Only "ACFA".

Ll. 91, 109, 182: "$N_d$", not "Nd".

L. 96: See Shaw et al. (1988) for what?

Fig. 1, l. 122: Define "Chl-a".

Fig. 1: What are "MOD" and "MYD"?

Ll. 128 – 129: Switch "upper" an "lower"?

L. 189: Define G18.

L. 213: Use "PP" instead of "primary production".

L. 269: "A1", not 1.

**References**

Freud, E., & Rosenfeld, D. (2012). Linear relation between convective cloud drop number concentration and depth for rain initiation. *Journal of Geophysical Research: Atmospheres*, *117*(D2).

Glassmeier, F., Hoffmann, F., Johnson, J. S., Yamaguchi, T., Carslaw, K. S., & Feingold, G. (2021). Aerosol-cloud-climate cooling overestimated by ship-track data. *Science*, *371*(6528), 485-489.

---

## Author Comment (AC1)

This short letter describes an analysis that combines MODIS satellite estimates of cloud droplet concentration in liquid-dominated marine low clouds with trajectory analysis over the Southern Ocean. The findings indicate that high concentrations of cloud droplets (Nd) tend to occur to the south (poleward) of a boundary previously identified as a "compositional front" that rings Antarctica. South of the "atmosphere compositional front of Antarctica (ACFA)" at roughly 60S comprises extremely biologically rich ocean waters that are copious sources of aerosol precursor gases (in particular dimethyl sulfide). Air mass back trajectories from high Nd clouds tend originate more frequently south of the ACFA. The high Nd south of 60S are associated with smaller effective radii and higher cloud optical thickness, but only marginally higher LWP, indicating that the cloud optical depth increase is largely driven by higher Nd, i.e., Twomey brightening.

The results presented here are interesting and important and I think very relevant to the ACP readership. I recommend publication subject to some minor revisions.

The main question I would like to raise is that I believe that the latitudinal gradient of light precipitation may also play an important role in setting the Nd latitudinal gradient through coalescence scavenging, in addition to the consideration of aerosol sources. We know from spaceborne 94 GHz radar that light precipitation maximizes at around 55S and decreases southward of this (see e.g., McCoy et al., 2020), so the reducing precipitation south of the ACFA may also be partly responsible for high Nd there. Another paper by Kang et al. (2022) illustrates the significant role that precipitation sinks may play. I wonder if the authors have tried to use any of the ship or aircraft measurements associated with CAPRICORN/MARCUS/SOCRATES to explore how precipitation sinks may change across the ACFA.

*Response:  There is no question that precipitation plays a key role in controlling the concentration of liquid cloud droplets in shallow boundary layer clouds.  We noted the role of drizzle in the original manuscript near line 220.  In the revised manuscript, we also add the important finding from Kang et al. and, we note also in the conclusions that the latitudinal gradient in Nd has many influences beyond a simple aerosol explanation.  Both Kang et al. (2022) and McCoy et al. (2020) are referenced in the revised manuscript.*

**Other points**

1. Line 35. Albedo increases with solar zenith angle, so how is this accounted for? Also, I didn't see any albedo measurements in the paper.

*Response: We have added CERES-derived albedo measurements in the revised manuscript. To account for the variation of cloud albedo with solar zenith angle, we developed and implemented an empirical correction for latitude using calculations presented in Minnis et al., 1998. The method for normalizing the CERES albedos for latitude are described in the methods section (Appendix) of the revised manuscript. The results of the albedo analysis are supportive of the original conclusions and make the paper much more compelling.*

2. Line 47-50: Why does a lack of precipitation make clouds more sensitive to CCN? Shiptracks in precipitating boundary layers tend to more visually apparent than those forming in non-precipitating clouds.

*Response: The analysis of Kang et al., 2022 that uses the simple empirical model of Wood et al. (2012) is based on the sources and sinks of Nd in liquid clouds. We reason that when precipitation is weak or absent, one of the sinks of Nd is also absent and, therefore, the sensitivity of Nd to CCN (a source of Nd) would be enhanced.*

3. Figure 1b does not seem important. Can't the essence of this simply be stated in the text?

*Response: Agree. Figure 1b has been removed and we have adapted the text to describe the main points.*

4. Line 95 and several studies point out the importance of air masses moving from interior Antarctica over the ocean as being the source of new particles. I do not understand why the Antarctic continent would be a good source of aerosol or aerosol precursor gases. It seems as though the highly productive ocean waters south of the ACFA are the main sources of aerosol. Can the authors comment on this?

*Response: I believe this statement to be accurate. The reason that trajectories that pass over Antarctica seem to have a higher CCN concentration is not yet definitively established in the observational literature. However, we know from prior studies that one of the pathways for nucleation of new aerosol and eventual growth to CCN from precursor gasses requires ultraviolet sunlight and that this process often happens in the free troposphere above low-level clouds. Air masses that pass over Antarctica with sufficient precursor gasses certainly get a higher dose of UV because of the high albedo of the ice-covered continent. The glaciated surface of Antarctica is also elevated and has low overall cloud cover. We speculate that trajectories passing over the elevated, high-albedo surface (especially in summer with long days) would encounter conditions conducive to new particle formation. We have attempted to suggest this process in the revised manuscript although we would like to be somewhat conservative in promoting the idea because of the lack of definitive observational evidence.*

5. Figure 1a: why not provide the correlation coefficients between cloud variables to make the points quantitatively?

*Response: McCoy et al. (2015) do an extensive regression analysis of the relationship between MODIS and factors responsible for Chl-a variations. Our results are consistent with their results. The correlation coefficients of the various quantities in Figure 1 are now noted in the revised manuscript.*

6. Line 116: LWP can remove aerosol, suppressing Nd (Wood et al., 2012). Nd can suppress precipitation, but the LWP response to this is bidirectional, and depends upon whether the background clouds were precipitating and up the dryness of the free troposphere. I don't think you can necessarily conclude that the seasonal cycle of LWP is dominated by meteorology (i.e. is NOT driven by aerosol, at least in part).

*Response: I don't think that LWP, per se, is the cause of $N_d$ change. Coalescence scavenging would increase as LWP (and likely $r_e$) increases. As $N_d$ increases, $r_e$ would decrease and LWP increase because of drizzle suppression. I assume that the entrainment drying of the MBL would tend to increase in summer as the free troposphere dries due to warming and a lower frequency of deep storms thereby decreasing LWP. All these factors are, of course, inter-related and saying that $N_d$ and $r_e$ are independent of LWP is hard to justify without a more thorough analysis that would be beyond the scope of this letter. So, I have softened the language here a bit and removed the statement that LWP is independent of $N_d$.*

7. Line 122: Provide evidence of the one month lag between Chl-a and Nd. Is this at all locations across the SO?

*Response: I say that this is the case in 4 of the 5 years. It can be seen by inspection of Figure 2 that Chl-a is rising about a month ahead of Nd (and decrease in re). Figure 2 is the result of averaging the MODIS and Chl-a retrievals over the entire analysis domain in each month. This result becomes much noisier when examined on finer spatial scales although, as shown by McCoy et al., (2015) for lower latitudes, there is a statistically significant relationship in broad regions of the Southern Ocean. A 1-month lag correlation increases the correlation between Nd and Chl-a from 0.27 to 0.60. However, because of the break in the time series (recall that we are examining November through February of each year) interpreting this quantitatively should be avoided. I think what the lag correlation captures is just what can be seen visually in the time series where Chl-a tends to rise about a month ahead of Nd in 4 of the 5 years.*

8. Fig 2/Line 134: This Nd gradient is documented and discussed in McCoy et al. (2020).

*Response.  True. Appropriately noted in the revised manuscript in the paragraph starting around line 200.*

9.  Line 156: Cite Korhonen et al. (2008), who established the pathway through the free troposphere. I would have expected the need for transport to the FT and nucleation of new particles to effectively reduce the sharpness of the Nd gradient driven by the gradient in surface-emitted precursor gases. Sources will lose their identity through the mid-deep tropospheric mixing and latitudinal displacement related to cyclonic systems. I would appreciate if the authors can comment on this issue. Line 179 seems to partly challenge the Korhonen transport pathway being primarily through the free troposphere.

*Response.  I did not mean to challenge the idea that transport of aerosol through the free troposphere is unimportant.  I do address the importance of new particle formation in the free troposphere and transport there in the paragraph around line 220.  While we do not address it here in detail because we only examine the summer, the entire Southern Ocean undergoes a large seasonal oscillation in CCN (Gras and Keywood, 2017) and Nd (McCoy et al., 2015; Mace and Avey, 2017) that has been documented although not fully explained.  It is my opinion (and only something of an hypothesis at this point) that CCN formed by new particle formation in the deep southern latitudes seeds much of the rest of the SO through northward transport through the free troposphere.  We hint at this in the last paragraph of the introduction and expand further upon it in the revised manuscript and in the conclusions.   Glen Shaw hints at this process in his early 1988 paper and again in 2007.  We also cite Korhonen et al., 2008 in the last paragraph of the introduction and again in the paragraph around line 220.*

10. Line 169-171: Are these 3D trajectories, or 2D? What method was used to determine the vertical ascent (model vertical velocity, isentropic....)?

*Response:  The HYSPLIT trajectory model is described in Stein et al. (2015).  The trajectory model uses the 3d model grids.  The vertical motions and horizontal winds are as predicted in the GDAS model.*

11. 4: The differences between the latitudes crossed by high Nd and low Nd trajectories shown here are quite modest yet are described as "overwhelming" (line 179). Does this statement pertain to clouds only south of the ACFA? It certainly does not pertain to high Nd cloud north of 60S since the majority of trajectories ending north of 60S never go below 60S.

*Response: No argument.  I have removed the "overwhelming" adjective.*

12. Line 245: No shortwave measurements are presented in the paper, so I'm not sure that the term "brightening" is appropriate unless said measurements are presented.

*Response:  We have added the CERES albedo in the revision to support the brightening claim.*

*All papers cited in our response are listed in the revision with the following exception.*

*Wood, R., Leon, D., Lebsock, M., Snider, J., & Clarke, A. D. (2012). Precipitation driving of droplet concentration variability in marine lowclouds. Journal of Geophysical Research, 117(D19). https://doi.org/10.1029/2012jd018305*

**References**

Kang, L., Marchand, R. T., Wood, R., & McCoy, I. L. (2022). Coalescence scavenging drives droplet number concentration in Southern Ocean low clouds. Geophysical Research Letters, 49, e2022GL097819.

Korhonen, H., Carslaw, K. S., Spracklen, D. V., Mann, G. W., & Woodhouse, M. T. (2008). Influence of oceanic dimethyl sulfide emissions on cloud condensation nuclei concentrations and seasonality over the remote Southern Hemisphere oceans: A global model study. Journal of Geophysical Research, 113(D15). https://doi.org/10.1029/2007JD009718

McCoy, I. L., McCoy, D. T., Wood, R., Regayre, L., Watson-Parris, D., Grosvenor, D. P., Mulcahy, J. P., Hu, Y., Bender, F. A.-M., Field, P. R., Carslaw, K. S., & Gordon, H. (2020). The hemispheric contrast in cloud microphysical properties constrains aerosol forcing. Proceedings of the National Academy of Sciences, 117(32), 18998–19006. https://doi.org/10.1073/pnas.1922502117

---

## Author Comment (AC2)

**Review of "Natural Marine Cloud Brightening in the Southern Ocean" by Mace et al. (acp-2022-571)**

The presented study analyzes strong gradients in the cloud droplet concentration found in the Southern Ocean, using five years of satellite observations. The authors show that these stark differences in the cloud microphysical composition can be traced back to biological primary production at the Antarctic Shelf, from where airmasses with high cloud droplet concentrations are moved to the north, while low cloud droplet concentrations originate from the open sea equatorward.

Despite many technical issues and a few minor comments, I enjoyed reading this manuscript. In a concise way, the article informs about the aerosol and cloud microphysics of an important region of the Earth. Thus, I support publication in *Atmospheric Chemistry and Physics* once my concerns are addressed. However, this article is also submitted to be published as an *Atmospheric Chemistry and Physics Letter*. In the current form, I cannot support the publication in this format, as I will outline in my only major comment below, but I am willing to be convinced otherwise.

**Major Comment**

*Does the article meet the requirements for an Atmospheric Chemistry and Physics Letter?* It is stated that an *Atmospheric Chemistry and Physics Letter* must fulfill the following requirements (see www.atmospheric-chemistry-and-physics.net/about/manuscript_types/acp_letters.html):

- Important discoveries and research highlights in atmospheric chemistry and physics.

- Solutions to or progress with long-standing and important questions in atmospheric research.

In its current form, the manuscript does not give substantial hints on how these requirements are fulfilled. I understand that the Southern Ocean is one of the least understood parts of the climate system, with inherent problems in modeling it. But how does the presented study contribute to improving its understanding? What are the important discoveries made? While I admit that the presented research is very interesting, the authors should use the opportunity to frame their work and highlight the advancements made through their work.

*Response: The manuscript builds upon prior research in several important ways that I think elevate the findings to the level of "important discoveries and research highlights". Prior work (D. McCoy et al., 2015) illustrate the correlations among various MODIS-derived cloud parameters and the processes associated with biogenic aerosol production while I. McCoy et al., (2020) documents the latitudinal variability in Nd in the Southern Ocean and claims that this ocean basin is the last vestige of the preindustrial Earth. In the present manuscript we find that the gradient in Nd associated with the production of biogenic aerosol in the high latitude Southern Ocean results in clouds that are significantly more reflective than their lower latitude counterparts. The higher albedo of these clouds with lower overall liquid water is a significant finding (discovery, if you will) that implicates biological processes in modulating the surface radiative balance of the high latitude Southern Ocean. While cloud property and aerosol sensitivities to biology have been documented in this region, a direct connection to radiative effects has not been documented until now. The higher albedos for lower liquid water path along the Antarctic Shelf now firmly establish that the surface solar radiation along the Antarctic shelf within the highly productive zone is modulated by the biology. This represents a forcing. The CLAW hypothesis (which we do not mention) describes a feedback – that the biology will change to keep the environment conducive to itself. The forcing that we identify is a necessary but not sufficient condition to establish the CLAW feedback.*

**Minor Comments**

Ll. 47 – 50: What cloud process is sensible to the CCN concentration? In non-precipitating clouds, an increase in CCN will not change the precipitation efficiency. I guess the authors refer to the cloud albedo.

*Response: Yes, something like the albedo susceptibility is what I refer to although we do not*

*quantify that derivative specifically in this study. While I think such a quantitative analysis would be interesting, it would put the present paper out of scope of a letter.*

Ll. 56 – 58: Define what "primary production" is. This will also help to frame the importance of the work (major comment). Do ll. 79 – 84 also refer to primary production?

*Response: Primary productivity refers to the net organic matter, mostly produced by phytoplankton, that is suspended in the ocean. The definition has been added to the text.*

Ll. 105 – 108: What is the purpose of this sentence? $N_d$ and $r_e$ are in opposite phase, not out of phase. So there is a very tight relationship.

*Response: I was simply pointing out how Nd varies with effective radius due to equation A1. Wording changed.*

L. 108: Replace "cycle" with "variability"?

*Response: Done.*

Fig. 2b: Add labels. The blue line represents locations with $N_d > 101$ cm$^{-3}$ and the green line $N_d < 42$ cm$^{-3}$. What about adding a red line for 42 cm$^{-3}$ < $N_d$ < 101 cm$^{-3}$?

*Response: We have clarified this in the caption. The colors of the histograms in the b-d are as defined in the inset of panel a.*

*Here is a plot with the middle tercile. I don't think it adds much to the discussion to include this since the histograms sit in between the upper and lower as one might expect.*

[Figure]

Ll. 113 – 117: The LWP changes non-monotonically with $N_d$. It increases with $N_d$ for precipitation stratocumulus, but decreases for non-precipitating stratocumulus due to increasing entrainment rates (Glassmeier et al. 2021).

*Response: Thank-you for pointing this out. I have mentioned this in the revision and cite the Glassmeier paper.*

Ll. 131 – 134: Why is $N_d$ only large at the ACFA and not below it?

*Response: I think the peak near 65S is just an artifact of the analysis. Figure 3 shows that the high Nd quartiles occurrence peaks immediately adjacent to Antarctica. However, when compiling the frequency distribution in Figure 2, the higher Nd occurrence at maximum latitude is minimized due to the decreased ocean surface area along the continental margin. I've removed this potentially confusing statement in the revision.*

Ll. 135 – 139: With a mean $r_e$ of 13 µm north of the ACFA, droplet coalescence might decrease $N_d$. See, e.g., Freud and Rosenfeld (2012), who showed that at 14 µm surface precipitation occurs, i.e., there is probably some drizzle at a slightly smaller $r_e$. This should be discussed.

*Response: I address this in the response to the comment below.*

Ll. 275 – 278: The significantly lower adiabaticity north of the ACFA could be due to precipitation, triggering the transition of closed- to open-cell stratocumulus. Discuss the possibility of precipitation.

*Response: I address the previous two comments together. The Freud and Rosenfeld paper analyzes in situ data collected in cumulus clouds. We are analyzing mostly stratocumulus clouds where the threshold relationships found in Freud and Rosenfeld might be different although it is likely that such a threshold exists. We do note in the revision the increased likelihood of precipitation in the lower latitude clouds given the larger droplet sizes and potential that precipitation is responsible for the lower adiabaticity of the lower latitude clouds.*

Ll. 297 – 300: Clouds with $r_e$ > 14 μm are usually precipitating (Freud and Rosenfeld 2012). Declaring all pixels with $r_e$ < 50 μm to non-precipitating clouds will cause a substantial bias. Please elaborate.

*Response: Our objective is to have cloud scenes that are mostly non precipitating (note we have changed the characterization to weakly precipitating in the methods section). A requirement for weakly precipitating clouds is because of the potential biases in cloud effective radius that occurs when pixels have significant precipitation water coexisting with a cloud droplet mode (Xu et al., 2022). We have found that the water path filter is the most important and that the effective radius criteria is irrelevant. To avoid confusion, I've removed reference to an upper effective radius bound.*

*Z. Xu, G. G. Mace and D. J. Posselt, "Impact of Rain on Retrieved Warm Cloud Properties Using Visible and Near-Infrared Reflectances Using Markov Chain Monte Carlo Techniques," in IEEE Transactions on Geoscience and Remote Sensing, vol. 60, pp. 1-10, 2022, Art no. 4110110, doi: 10.1109/TGRS.2022.3208007.*

**Technical Comments**

*Response: All Technical comments addressed as suggested*

L. 32: "Both" usually refers to two objects. Here, it refers to three (latitudinal, longitudinal, and temporal). Revise.

L. 45: SO for Southern Ocean is already defined. Use it.

L. 55: To what is "respectively" referring to?

L. 66: Why is "TOF" defined? It is never used.

L. 88: "ACF" is not defined. Only "ACFA".

Ll. 91, 109, 182: "$N_d$", not "Nd".

L. 96: See Shaw et al. (1988) for what?

Fig. 1, l. 122: Define "Chl-a".

Fig. 1: What are "MOD" and "MYD"?

Ll. 128 – 129: Switch "upper" an "lower"?

L. 189: Define G18.

L. 213: Use "PP" instead of "primary production".

L. 269: "A1", not 1.

**References**

Freud, E., & Rosenfeld, D. (2012). Linear relation between convective cloud drop number concentration and depth for rain initiation. *Journal of Geophysical Research: Atmospheres*, *117*(D2).

Glassmeier, F., Hoffmann, F., Johnson, J. S., Yamaguchi, T., Carslaw, K. S., & Feingold, G. (2021). Aerosol-cloud-climate cooling overestimated by ship-track data. *Science*, *371*(6528), 485-489.

---

## Author Response (AR2)

==============================

l. 57: the paper by Petters and Kreidenweis, 2007, describes a parameterization of CCN hygroscopicity. It does not seem to be a correct reference in the context of the role of CCN concentrations on shallow boundary layer clouds.

Response:  Removed Petters and Kreidenweis and added Painemal et al. 2017.

l. 83: replace 'aerosol chemistry' by 'chemical composition of aerosol' or 'chemical processes'

Response:  Done

l. 101: what do you mean by 'sulfur-based concentrations of aerosol'? Do you mean 'highest sulfate masses' or 'highest particle number concentrations of sulfur-containing particles' or something entirely different?

Response:  This is the language used in Twohy et al., (2021) that defines sulfur-based in their supplemental text as measurements from the STEM-EDS instrument during Socrates as  "Round shape, primarily S, O, may be volatile under the electron beam. (Sulfuric acid, ammonium sulfate/bisulfate or MSA)". I changed the language to be more specific of what I think Twohy et al. is referring to – basically non sea salt sulfates.

l. 121: (1) what exact relationship was used? What was different to the one by Stephens (1978) cited here? (2) Please write the equation in a separate line and add an equation number.

Response.  The stated equation is the relationship used.  I re-wrote to be more precise and added an equation number as you requested.

l. 123 – 5: This sentence seems awkward. It might be clearer "... with a correlation coefficient of -0.60 in the monthly means (Figure 1)."

Response:  Yes, the sentence was awkward.  Changed as suggested.

l. 175: 'deposition' is not a chemical pathway. Maybe better: 'Other pathways are possible such as condensation of sulfates onto primary sea salt particles... '

Response:  Changed as suggested.

l. 177: It is not clear what you mean by 'removal of sulfur compounds' – even after

oxidation, they remain sulfur compounds. Or do you mean 'uptake and aqueous oxidation of sulfur-containing gases'.i.e. removal from the gas phase?

Response:  My chemistry credentials are not strong, obviously.  I changed the language to represent what Woodhouse et al is implying.  As I understand it, they argue that gas phase sulfur compounds are removed from the gas phase through aqueous phase oxidation in cloud drops.

Figure 1 is the key figure of the manuscript. Please make sure that at least this figure is accessible for readers with color vision deficiencies (cf Figures & Tables at https://www.atmospheric-chemistry-and-physics.net/submission.html#manuscriptcomposition) . For example, choose a different color scheme or split the figure into two panels using the same x-axis or make the lines better distinguishable, independent of their color by using different line types/strengths or symbols.

Response:  Figure 1 is now in two panels, and we use red and black with different line styles and symbols.

Throughout the manuscript: Please write 'd' as index in Nd according to your definition in l. 63.

Response:  Found three instances and fixed those.